# Spectro-temporal acoustical markers differentiate speech from song across cultures

Philippe Albouy [1,2,3] ✉, Samuel A. Mehr [2,4,5], Roxane S. Hoyer[1], Jérémie Ginzburg[1,6,7], Yi Du [8] & Robert J. Zatorre [2,3,7] ✉

Humans produce two forms of cognitively complex vocalizations: speech and song. It is debated whether these differ based primarily on culturally specific, learned features, or if acoustical features can reliably distinguish them. We study the spectro-temporal modulation patterns of vocalizations produced by 369 people living in 21 urban, rural, and small-scale societies across six continents. Specific ranges of spectral and temporal modulations, overlapping within categories and across societies, significantly differentiate speech from song. Machine-learning classification shows that this effect is cross-culturally robust, vocalizations being reliably classified solely from their spectro-temporal features across all 21 societies. Listeners unfamiliar with the cultures classify these vocalizations using similar spectro-temporal cues as the machine learning algorithm. Finally, spectro-temporal features are better able to discriminate song from speech than a broad range of other acoustical variables, suggesting that spectro-temporal modulation—a key feature of auditory neuronal tuning—accounts for a fundamental difference between these categories.

Human vocal communication involves two distinct modes: speech and song[1–4]. Unlike simpler vocalizations (screams, laughter, etc.), singing and speaking rely on temporally extended recursive grammatical structures and are based on arbitrary mappings that must be learned within each culture[5]. A great deal of work has documented the remarkable cross-cultural variability in both the structural features of speech and song, as well as their acoustical manifestations[6–12], but debate continues about whether the two categories may be distinguished across societies on the basis of acoustical features alone. Speech and song are produced by the same vocal tract, yet each makes distinct demands on musculature, breathing, and motor control mechanisms[13,14], raising the possibility that certain acoustical cues could serve as markers of each category[15]. However, even though

people readily distinguish speech and song, the cues underlying the categories, even within cultures, are far from clear[11,16–19], so that such a claim is difficult to address. Indeed, even if speech and singing reliably exist as separate, recognizable entities, their cognitive representation could depend mostly on learned regularities that are particular to each cultural group.

One source of difficulty in comparing speech and song is that they each form part of broader communication systems of language and music, respectively. These systems share certain features (e.g., syntax), but also differ in important ways (e.g., the hierarchical organization of metrical patterns)[5,20]. Pitch variation in music tends to be more discrete than in speech[1,12,21], leading to the formation of hierarchical tonal organization[22], which may be a fundamental property of music

[1]CERVO Brain Research Centre, School of Psychology, Laval University, Québec City, QC, Canada. [2]International Laboratory for Brain, Music and Sound Research (BRAMS), Montreal, QC, Canada. [3]Centre for Research in Brain, Language and Music and Centre for Interdisciplinary Research in Music, Media, and Technology, Montréal, QC, Canada. [4]School of Psychology, University of Auckland, Auckland 1010, New Zealand. [5]Child Study Center, Yale University, New Haven, CT 06511, USA. [6]Lyon Neuroscience Research Center, CNRS, UMR5292, INSERM, U1028 - Université Claude Bernard Lyon 1, F-69000 Lyon, France. [7]Cognitive Neuroscience Unit, Montreal Neurological Institute, McGill University, Montreal, QC, Canada. [8]Institute of Psychology, Chinese Academy of Sciences, Beijing, China. ✉e-mail: philippe.albouy@psy.ulaval.ca; robert.zatorre@mcgill.ca

worldwide[8]. But it remains unclear whether such descriptive differences represent acoustic phenomena invariant enough to form a sufficient basis for categorization across different musical and linguistic systems, or, instead, are merely associated with the two domains. And since the song has mostly been explored only in Western cultures it is uncertain if the features described to date are widely characteristic of most human songs or only of those that have been well studied[23].

Recent developments in neurophysiology and cognitive neuroscience offer a rigorous framework for testing how speech and song could differ. Complex sounds can be characterized according to the distribution of their spectro-temporal modulation (STM) power[24]. Neurons in auditory regions across various species can be described in terms of their spectro-temporal receptive fields, which have been shown to constitute an efficient coding scheme for complex acoustical patterns[25–27]. Spectral and temporal modulation dimensions are largely separable in both humans and monkeys[28]. Moreover, neuronal spectro-temporal tuning functions correspond well to the most relevant acoustical features that characterize different animals' communicative signals, including birdsong[27], cat meows[29], and monkey calls[30], indicating a match between the acoustics of important sounds in the environment and the neural hardware needed to process them.

Might spectro-temporal modulation content constitute a fundamental, and sufficient difference to account for how speech and song differ from one another? Acoustical analysis shows that speech tends to contain faster temporal modulations than music[12,15,31], and temporal modulation cues are well-known to be sufficient for speech perception, even when spectral modulations are degraded[32]. Conversely, degradation of spectral modulations abolishes the perception of melodic content in song, while leaving speech comprehension intact, whereas degradation of temporal modulations renders the speech content of songs incomprehensible but has little effect on the melody[33]. These findings dovetail well with the idea that spectral and temporal features are processed in partially distinct neural populations within[34,35] and across the two hemispheres[33,36,37].

Taken together, those results suggest that speech and song may exploit different ends of the spectro-temporal continuum. But such a conclusion suffers from a major limitation because although the high temporal rate of speech has been confirmed for many distinct languages[38], the spectro-temporal features of music have mostly been characterized in a limited Western musical repertoire, which is not necessarily representative of all human musical systems. Whether the role of spectro-temporal modulations in distinguishing speech from song is an idiosyncrasy of some cultures, or whether it represents a more fundamental aspect of the biology of human communication – as one would expect, given the fundamentally different functional roles of speech[39,40] and music[2] in human evolution, and their partly distinct neural representations – is the question we address in this paper.

Specifically, we tested whether distributions of STM power in speech and song are sufficient to distinguish the two vocalization types within and across 21 societies sampled from all inhabited continents and comprising small-scale, rural, and urban societies. The recordings, produced in 18 languages from 12 language families, were gathered from native speakers of each language who each lived in the society where the recording was gathered (see ref. 6 for full details and Table S1 and Fig. S1). Three hundred sixty-nine people from these societies were asked (i) to speak in a casual, ordinary fashion, on a mundane topic directed to the experimenter (e.g., describing their daily routine); and (ii) to sing a song of their choice, with the only requirement being that the song was not intended to be sung to an infant. Whether the vocalization was considered to be an example of speech or song was therefore determined strictly by the person producing the vocalization, and not imposed in any way by the researcher. Importantly, we analyzed only matched pairs of speech and song

produced by the same individual, allowing us to test for category difference independently of individual characteristics of each person's voice (such as pitch, breathiness, nasality, etc.), which would otherwise confound the comparison.

We predicted (i) that if speech and song are characterized by distinct STM signatures, we should be able to observe distinct distributions of these patterns with appropriate acoustical analysis; (ii) that if such differences are truly common across societies, we should observe substantial overlap in the distribution of STM power for each category (speech, song) across all societies studied; (iii) that if these STM markers are sufficient to categorize the two classes of vocalizations, then a machine-learning classifier should be able to determine which sample corresponds to speech or song with adequate accuracy, based solely on their STM profile; (iv) that the information most used by the classifier should correspond to the spectro-temporal signatures derived from the initial acoustical analysis; (v) that listeners unfamiliar with the language or music of the different societies should nevertheless be able to correctly classify speech and song, with a similar ordering of accuracy across samples as the machine-learning classifier, if human judgments are based on spectro-temporal cues; and (vi) that if spectro-temporal modulation features constitute a fundamental, and sufficient difference to account for how speech and song differ from one another, classification performed with spectro-temporal features should be more accurate than decoding performed with a broad range of other acoustical variables (pitch, formants, intensity, vowel rates, rhythmic measures etc.).

In this work, we show that specific ranges of spectral and temporal modulations differentiate speech from song in a consistent fashion and that those ranges overlap within categories and across societies. Machine-learning analyses confirm that this effect is cross-culturally robust, with vocalizations reliably classified as song or speech solely from their spectro-temporal modulation patterns across all 21 societies. Listeners unfamiliar with the cultures could classify these vocalizations with similar accuracy patterns as the machine-learning algorithm, using similar spectro-temporal cues to those used by the classifier. Finally, we show that spectro-temporal features are better able to discriminate song from speech than a broad range of other acoustical variables, suggesting that spectro-temporal modulation content accounts for a fundamental difference between speech and song, beyond general acoustic cues.

## Results

We decomposed the acoustical signal of the vocalization samples using the Spectro-Temporal Modulation (STM) framework (Fig. 1). STM patterns for singing and speaking samples were extracted (ModFilter algorithm[41]) for each vocalization (see "Methods" section and ref. 33 for a similar procedure), and then used for univariate and multivariate analyses. The identical pipeline was used for both speech and song samples, thus avoiding any kind of bias in the procedure.

We contrasted the spectro-temporal modulation patterns of song and speech vocalizations using nonparametric permutation statistics with FDR correction in the spectral and temporal domains (as implemented in FieldTrip[42] and incorporated in Brainstorm[43]- see "Methods" section). This analysis revealed two hotspots of increased spectral modulations in song as compared to speech samples ($10^5$ permutations, FDR corrected, $p < 0.001$): hotspot 1: peak at 3.53 cyc/kHz in the spectral domain and 0.66 Hz in the temporal domain; hotspot 2: peak at 7.11 cyc/kHz in the spectral domain and −0.66 Hz in the temporal domain (note that human speech is symmetric between positive and negative temporal modulation frequencies[41]), which correspond to increasing and decreasing frequency trajectories, respectively.

We also detected three hotspots of increased temporal modulation in speech as compared to song: hotspot 1: peak at 6.16 Hz in the temporal domain and 0 cyc/kHz in the spectral domain; hotspot 2:

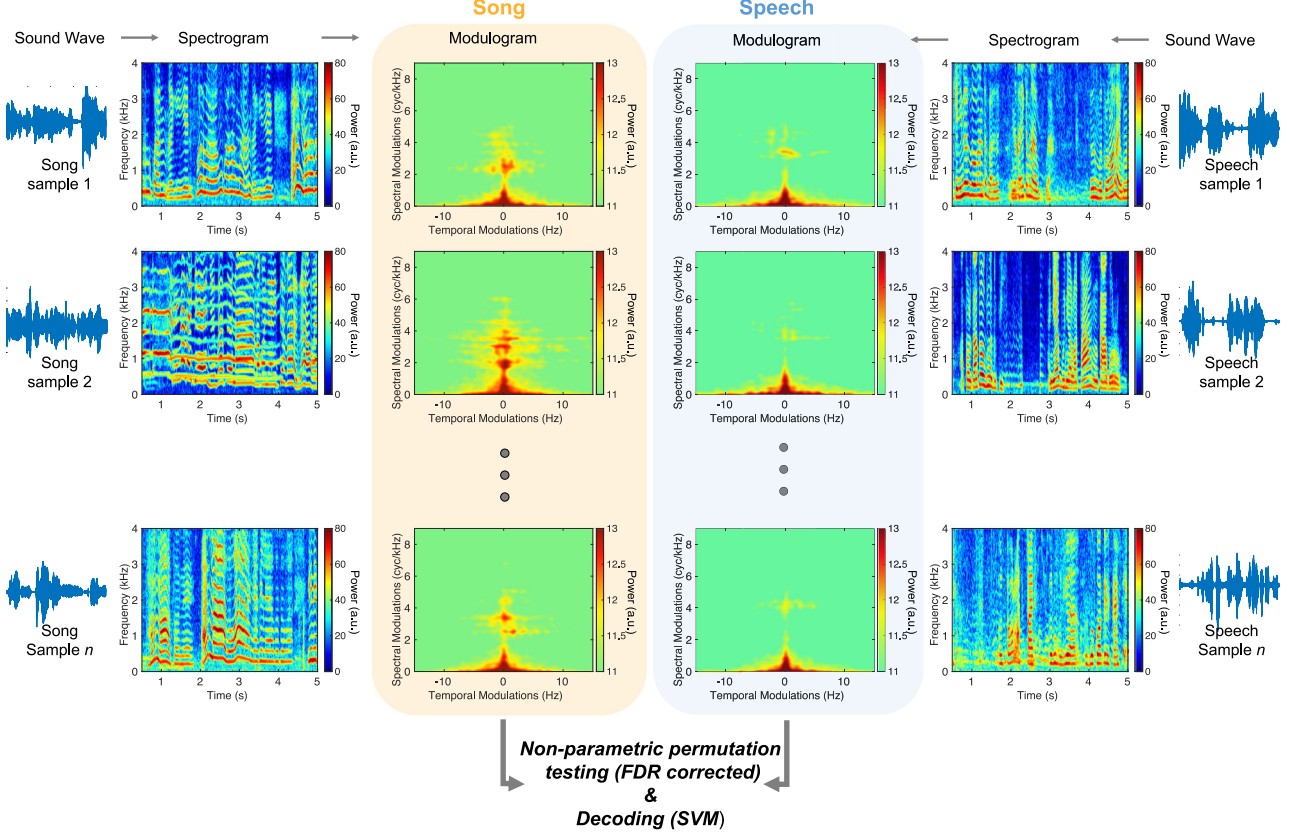

**Fig. 1 | Extraction of STM patterns for singing and speaking vocalization samples.** Sound waves, spectrograms, and modulograms of representative vocalizations (here, in recordings from the Nyangatom of Ethiopia; left panel: song, right panel: speech) revealing the acoustic complexity of the song and speech samples. For each sample, we extracted the STM (modulograms) patterns. We then contrasted the song and speech STM patterns using nonparametric permutation statistics (FDR corrected) and used the STM patterns data as features to perform a 2-class SVM decoding of music and speech samples (see Fig. 2).

peak at −6.33 Hz in the temporal domain and 0 cyc/kHz in the spectral domain; hotspot 3: peak at 4.83 Hz in the temporal domain and 5.07 cyc/kHz in the spectral domain.

To assess the consistency of this effect across societies we generated a heatmap illustrating the overlap in a number of societies that display a significant effect in the hotspots identified in Fig. 2A This analysis (Fig. 2B) revealed that 20/21 societies showed a significant increase of spectral modulations in singing samples vs speech samples at 3.71 cyc/kHz (in the spectral domain) and 0.66 Hz (in the temporal domain). Moreover, 20/21 societies showed a significant increase in temporal modulations in speech samples vs singing samples at 6.33 Hz (in the temporal domain) and 0.09 cyc/kHz (in the spectral domain). The robustness of this effect was also confirmed with a k-means clustering analysis performed on the coordinates in the spectro-temporal domain of the statistical peaks of each society for the contrast song vs. speech. Note that for this analysis the absolute values of temporal modulations were used, as human vocalizations are symmetric between positive and negative temporal modulation[41]. This analysis revealed 2 clear clusters with centroids at: cluster 1: 3.48 cyc/kHz (in the spectral domain) and 0.13 Hz (in the temporal domain, Fig. 2C) and cluster 2: 0.33 cyc/kHz (in the spectral domain) and 6.33 Hz in the temporal domain.

To confirm the cross-cultural robustness of these effects, we then used a Support Vector Machine (SVM) classifier with fieldsite-wise k-fold cross-validation to classify song and speech vocalization samples, using only the STM patterns as input features (see "Methods" section). This approach provides a strong evaluation of cross-cultural regularity: the model is trained only on data from 20 of the 21 societies to predict whether each vocalization in the 21st society is song or speech. The procedure is repeated 21 further times, with data from each society being successively held out, to estimate the classification performance across the full set of societies. The summary of the SVM's performance (average of all models) reflects, corpus-wide, the degree to which song and speech STM patterns are stereotyped because high classification performance can only result from high cross-cultural regularities.

The models significantly classified song and speech above chance (Wilcoxon rank test, two-tailed, $W(20) = 231$, $p < 0.001$; Rank biserial correlation (effect size) = 1.00; 95% Confidence Interval = [29.9 40.6]; Fig. 2D; accuracy = 84.5% ± 10.4 (SD); sensitivity = 83.8% ± 15.9, specificity = 85.2% ± 13.8; ROC curves for each society are presented in Fig. 2E). Evaluating classification performance within the recordings in each fieldsite showed a high degree of cross-cultural regularity, with the performance in all 21 fieldsites above chance level (Fig. 2D, E), even though accuracy varied across different sites. It is relevant to note that this variability in the performance of the model across societies could be explained by the sound sample duration: sound duration directly affects the quality of the STM estimation, and it also, in consequence, affects decoding accuracy (see the positive correlation $r(20) = 0.63$, $p < 0.001$ between SVM decoding accuracy against average sample duration (s) presented in Fig. S2).

We then investigated what STM features the model relied upon to discriminate song and speech STM patterns. For each classifier, we extracted the feature weights to estimate their relative importance (z-scored, averaged across societies). We identified four spectrotemporal patterns showing substantial differences in the features the model relied upon to reliably classify speech and song across societies (Fig. 2F). Furthermore, the four regions of the STM space most critical to the classifier's performance correspond well to the acoustical

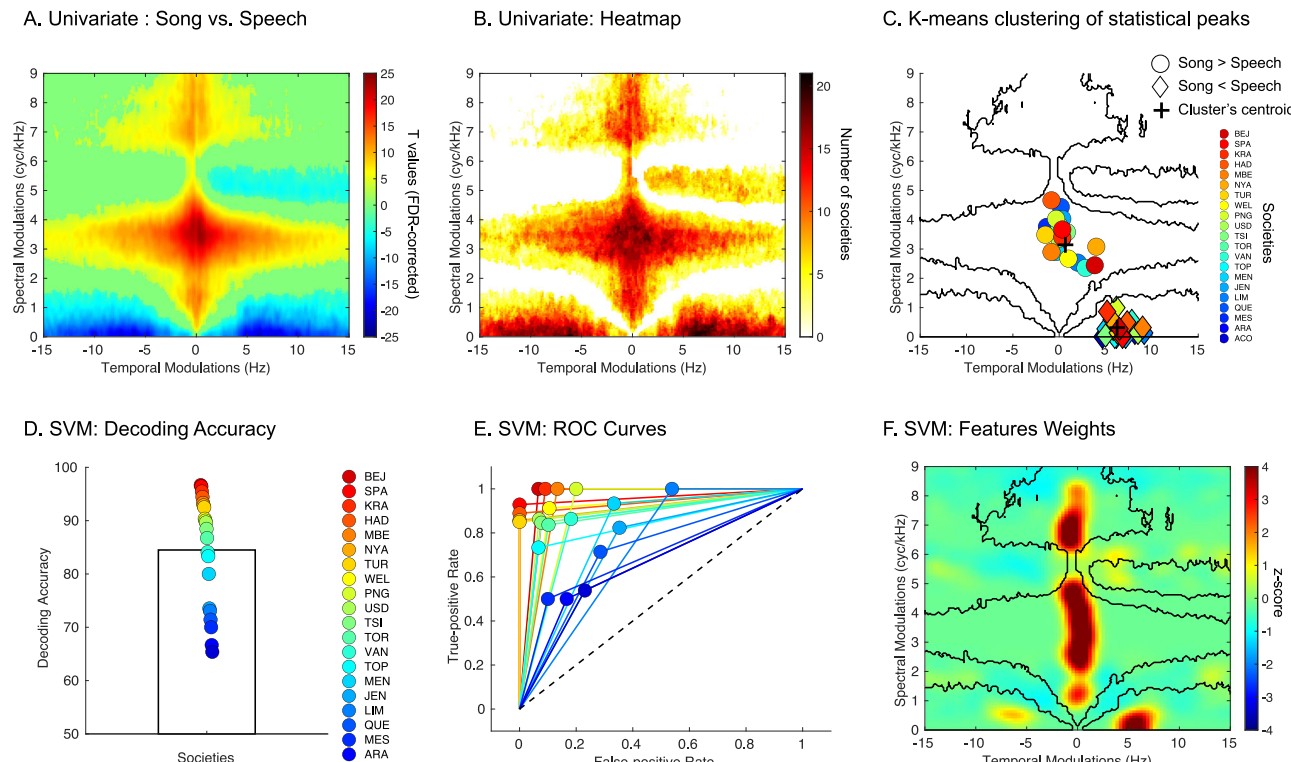

**Fig. 2 | Cross-cultural spectro-temporal markers of song vs. speech identified with univariate analyses and machine learning. A** Song vs. Speech contrast (two-tailed) in the STM domain across all societies ($p < 0.001$, FDR corrected in the spectral and temporal modulation domains, $n = 369$ independent vocalizations). **B** Heatmap (smoothed) depicting the number of societies showing a significant effect in the clusters identified in (**A**). Each value reports a numeric count, with larger counts associated with black coloring. **C** K-means clustering of statistical peaks; dots represent each society. Dark lines illustrate the boundaries of the significant effects presented in (**A**). **D** Fieldsite-wise cross-validated support vector machine decoding accuracy (chance level: 50%). The colored dots represent the accuracy for each society (sorted as a function of accuracy with a jet colormap) $n = 21$ independent societies. **E** Receiver operating characteristic curve (ROC) for each society (same color code as in (**A**). Black dashed line represents the chance level. **F** Normalized feature weights in the modulation power spectrum domain showing features with the largest influence (z-score, average of the 21 classifiers) for the classifier. Dark lines illustrate the boundaries of the significant effects presented in (**A**).

differences identified in the initial analyses (Fig. 2A, B) as shown by the *a posteriori* overlap observable in Fig. 2C, F.

To confirm the reliability of these findings, and to verify that the accuracy rates were not inflated by any incidental similarities between the samples used for cross-validation, we repeated the same analysis with four alternative cross-validation strategies, using the same cross-validation procedure but doing so across countries, language families, world subregions, and world regions instead of fieldsites (societies). The results robustly replicated in all cases with large effect sizes (Fig. 3, see Supplementary Material for detailed statistics).

**Behavioral analysis**

We then studied naïve listeners' sensitivity to these spectro-temporal features. We played the song and speech recordings to 74 individuals who were asked to rank, as rapidly as possible on a 5-point scale, whether each speaker was singing (code 1) or speaking (code −1) (see Fig. 4A). These primarily French-speaking listeners from Quebec (Canada) and France were presumably unfamiliar with the languages or music of most of the societies from which the sounds were recorded. Their judgments were accurate, with large effect sizes for both Song (Wilcoxon rank test, two-tailed, $W(73) = 2775$, $p < 0.001$; Rank biserial correlation (effect size) = 1.00; 95% Confidence Interval = [86.2 89.8]) and Speech (Wilcoxon rank test, two-tailed, $W(73) = 0$, $p < 0.001$; Rank biserial correlation (effect size) = 1.00; 95% Confidence Interval = [−92.7 −88.8]- Fig. 4B).

To test whether these listeners were using the spectro-temporal cues that distinguished song from speech in the prior analyses, we tested if the features identified on the STM patterns (see Figs. 2 and 3) could predict their behavioral ratings. To do so, we computed the normalized difference between Song STM and Speech STM and between Song and Speech behavioral ratings (with a positive score representing a large difference between song and speech ratings) for each of the 369 vocalizations/speakers (see "Methods" section). We then computed the correlation (FDR corrected, $p < 0.05$, Fig. 4D) between these difference scores and observed (i) a positive relationship between increased spectral modulation for song relative to speech (−0.33 Hz in the temporal domain and 3.35 cyc/kHz in the spectral domain – see Fig. 4D) and positive behavioral difference scores (corresponding to large difference ratings between song and speech) and (ii) a negative relationship between decreased temporal modulation for song relative to speech (4.49 Hz in the temporal domain and 0.09 cyc/kHz in the spectral domain) and positive behavioral difference scores (corresponding to large difference rating between song and speech– see Fig. 4D).

To test the consistency of our listeners' inferences across cultures, we computed the fieldsite-level behavioral ratings. Within each of the 21 societies, listeners' judgments were accurate, again with large effect sizes, for both Song (Wilcoxon rank test, two-tailed, $W(20) = 231$, $p < 0.001$; Rank biserial correlation (effect size) = 1.00; 95% Confidence Interval = [76.5 90.3]) and Speech (Wilcoxon rank test, two-tailed, $W(20) = 0$, $p < 0.001$; Rank biserial correlation (effect size) = 1.00; 95%

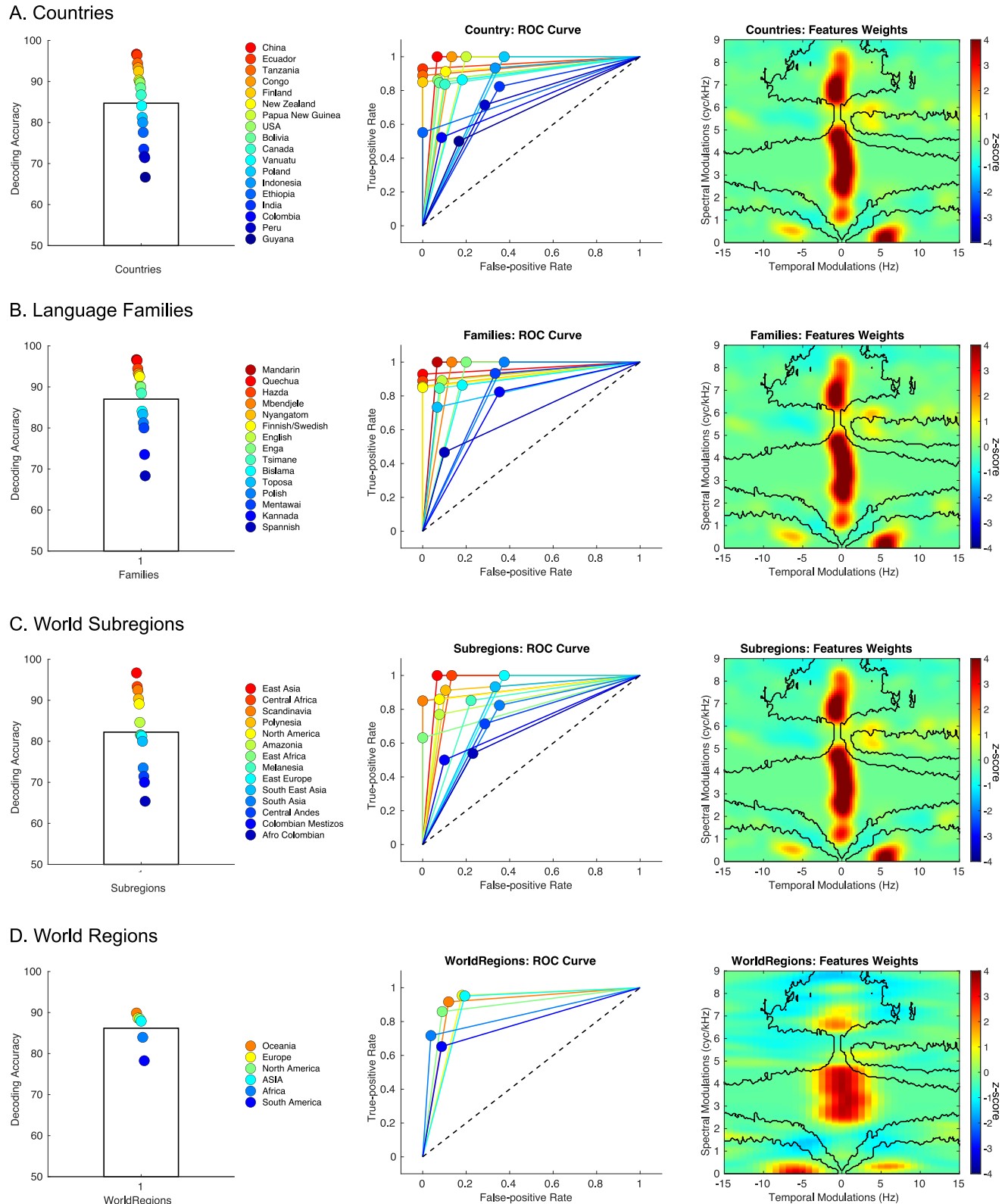

**Fig. 3 | Cross-cultural regularities across countries, language families, world subregions, and world regions identified with machine learning. A** Left Panel: Country-wise cross-validated decoding accuracy (chance level – 50%). The colored dots represent the performance accuracy for each country (sorted as a function of accuracy with a jet colormap) $n = 18$ independent countries. Middle Panel: Receiver operating characteristic curve (ROC) for each country (same color code as in the left panel). Black dashed line represents the chance level. Right Panel: Features weights in the MPS domain showing features with the largest influence ($z$-score, average of the 18 classifiers). **B–D** Same as (**A**) for language families ($n = 15$ independent families), world subregions ($n = 14$ independent subregions), and world regions ($n = 6$ independent regions) respectively.

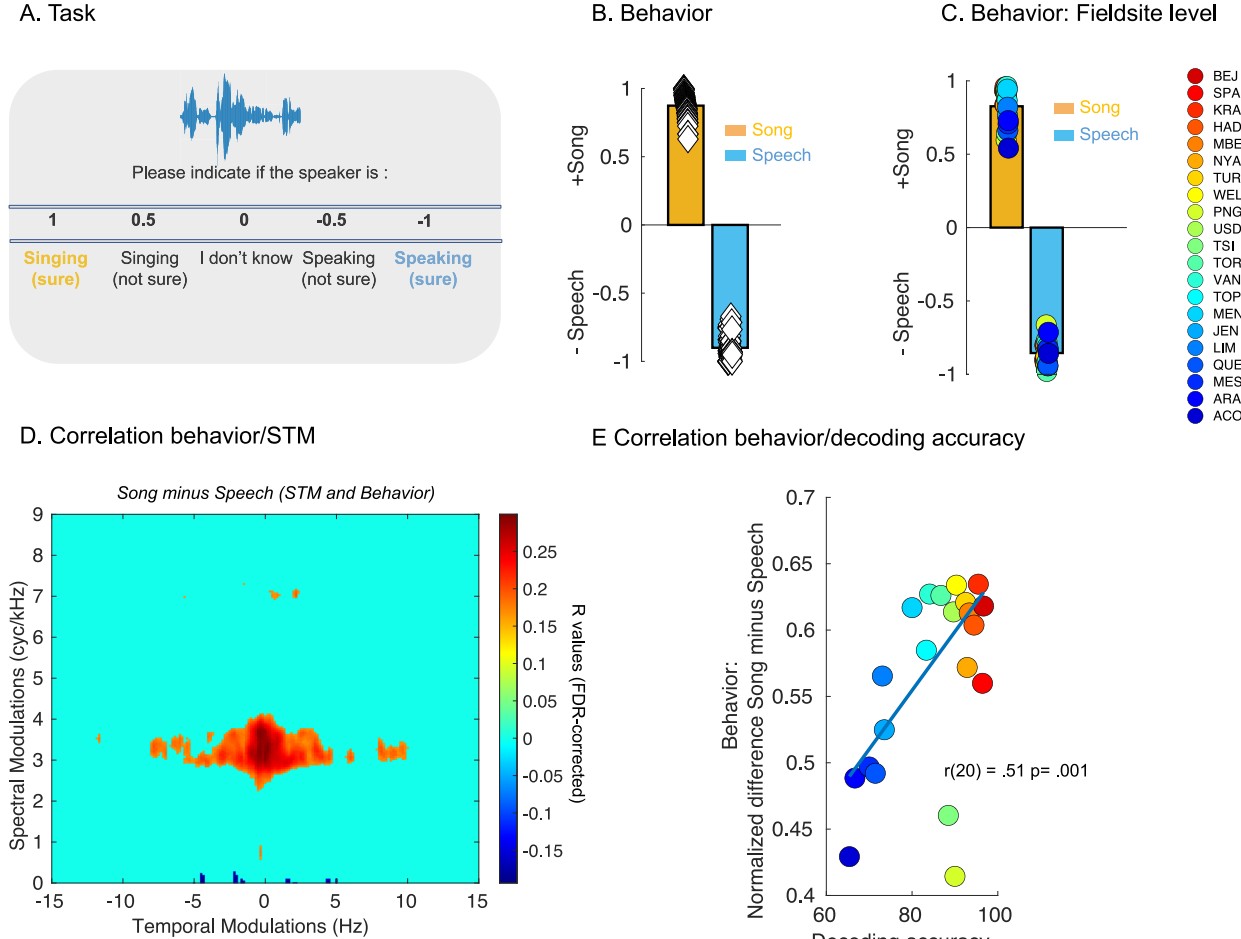

**Fig. 4 | Naïve listeners distinguish song from speech vocalizations across cultures. A** Behavioral task: 74 individuals were asked to rank, as rapidly as possible on a 5-point scale, whether each speaker was singing (code 1) or speaking code (−1), only 74 participants completed the experiment. **B** Behavioral ratings (chance level – 0) for song (orange) and speech (blue) samples. Diamonds represent the ratings for each listener ($n = 74$ independent individuals). **C** Fieldsites-level behavioral ratings (chance level – 0) for song (orange) and speech (blue) samples. Colored circles represent each of the 21 societies/cultures (sorted as a function of the SVM decoding accuracy of Fig. 2D - with a jet colormap - $n = 21$ independent societies). **D** Correlation between normalized difference scores (Song MPS vs. Speech MPS and Song vs. Speech behavioral ratings) represented in the MPS domain. (FDR corrected in the spectral and temporal modulation domains, $p < 0.05$). **E** Scatter plot of SVM decoding accuracy (Fig. 2D) against behavioral normalized difference (Song vs. Speech). Colored circles represent each of the 21 societies/cultures (sorted as a function of the SVM decoding accuracy of Fig. 2D, two-tailed).

Confidence Interval = [−89.9 −81.1]) - Fig. 4C, see Fig. S3 for the same analysis for countries, language families, world subregions and world regions.

Finally, to confirm that human judgments were based on similar spectro-temporal cues as those identified in the STM, we investigated whether these listeners unfamiliar with the different societies were identifying speech and song samples with a similar ordering of accuracy across samples as the machine-learning classifier (see Fig. 2). To do so, we computed the correlation between SVM decoding accuracy and the normalized difference between Song and Speech behavioral ratings computed within each society. As expected, this analysis revealed that decoding accuracy of the classifier was positively correlated with the normalized behavioral scores ($r(20) = 0.51$, $p = 0.001$, Fig. 4E).

**Acoustical Analysis**
Finally, we studied whether spectro-temporal modulation features constitute a fundamental, and sufficient difference to account for how speech and song differ from one another, or whether acoustic features

of the vocalizations might account for the results just as well. To do so, we used a broad range of acoustical variables extracted in ref. 6 (such as pitch (f0), first formant (f1), amplitude (intensity), pitch space, vowel rate, vowel space, roughness etc. see Supplementary Material and Table S2 for details) to test whether these variables were: (i) correlated with spectro-temporal features, (ii) could decode speech and song vocalizations with similar, higher or lower accuracy than with STM features, and (iii) whether decoding accuracy for acoustical data can predict the behavioral scores of naïve listeners.

To analyze relations between vocalization type, STM features, and acoustic features over and above the known correlations between STM features and acoustic features, we used Partial Least Squares (PLS) analysis, with the acoustic features as predictor variables and STM features as response variables. We used PLS instead of multiple linear regression models to take into account the multicollinearity in the acoustic features and applied the analysis to each spectral/temporal coordinate of the entire STM domain across song and speech, yielding in one PLS model per spectral/temporal coordinate. For each model we then extracted the fitted responses values and estimated if the

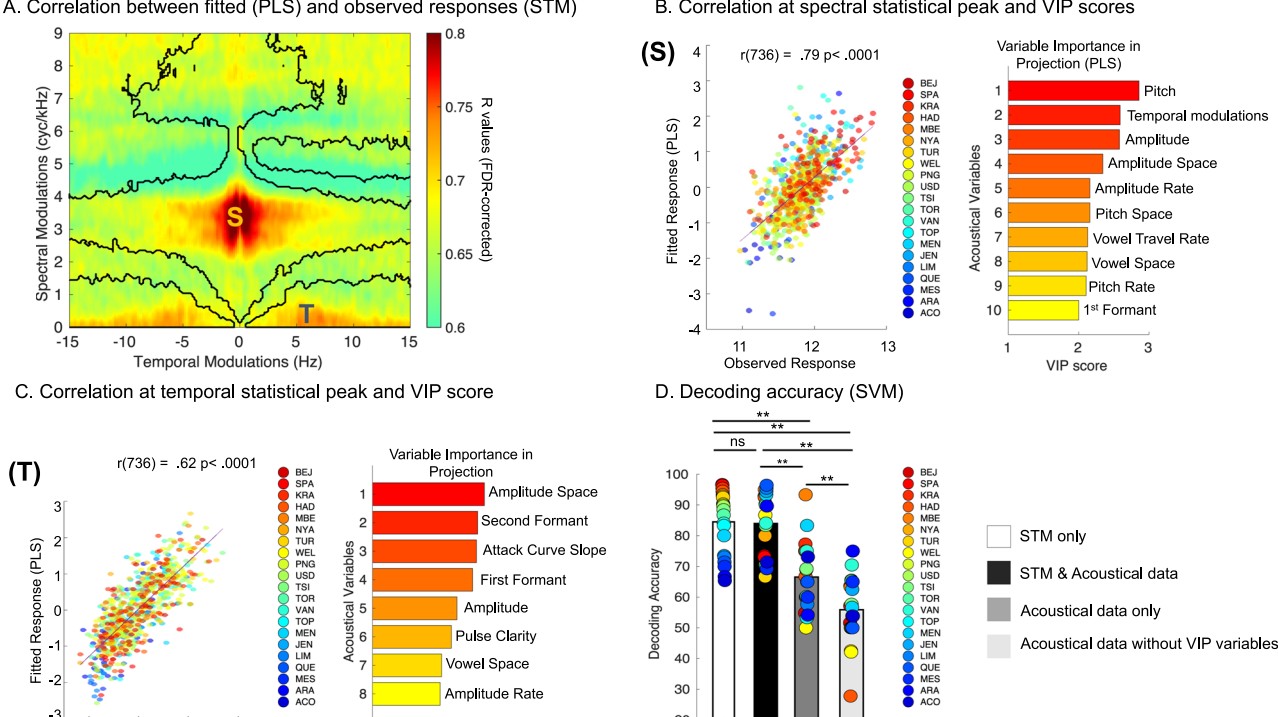

**Fig. 5 | Superiority of STM features over other acoustical variables. A** R values in the STM domain for the correlation between fitted responses (PLS) and observed responses (STM) (FDR corrected in the spectral and temporal modulation domains, $p < 0.05$, two-tailed). Dark lines illustrate the boundaries of the significant effects presented in (Fig. 2A). **B**, **C** Left Panels: Scatter plot of fitted responses (PLS) and observed responses (STM) for the spectral (see letter S in (**A**), statistical peak (**B**), and temporal (see letter T in (**A**) statistical peak (**C**)). Circles represents each speakers/vocalization ($n = 369$), two-tailed, all ps < 0.001. Right panels: VIP scores: the horizontal bars show the acoustic features with the largest influence in the PLS. **D** Fieldsite-wise cross-validated ($n = 21$ independent societies) support vector machine decoding accuracy (chance level: 50%) for four alternative strategies with STM features only (white), STM + acoustical features (black), Acoustical features only (gray) and Acoustical features without VIP variables (light gray). The colored dots (jet colormap sorted as a function of accuracy for the model trained with STM features only) represent the accuracy for each society. **p < 0.001, ns non-significant, post hoc pairwise comparisons were two-tailed, Bonferroni corrected.

models could significantly fit the STM data. To do so, we computed the correlation (FDR corrected, $p < 0.001$, Fig. 5A–C) between the fitted responses (PLS models) and observed response (STM features) in the entire STM domain.

This analysis revealed two clear hotspots which overlapped directly with the hotspots reported in Figs. 2, 3 and 4, where STM data is predicted by acoustical data (hotspot 1: peak at 3.49 cyc/kHz in the spectral domain and 0.49 Hz in the temporal domain, Fig. 5A, B; hotspot 2: peak at 0.13 cyc/kHz in the spectral domain and 7.13 Hz in the temporal domain, Fig. 5A, C). To determine which acoustical variables accounted most for these predictions, we calculated the variable importance in projection (VIP) scores for the PLS regression models for spectral and temporal statistical peaks reported in Fig. 5A; such VIP scores summarize the degree of contribution a variable makes to the model (see "Methods" section for further information). As variables with a VIP score greater than 1 are considered important for the projection of the PLS regression model[44], we report here only variables with VIP scores greater than 1. For the spectral hotspot, variables such as Pitch (fundamental measure of the highness or lowness, in frequency), Amplitude Space (dynamic measure of intensity's range over time), and Temporal Modulations were the variables that contributed most to the prediction of the STM data. For the temporal peak, Amplitude Space, and Second Formant were the variables that contributed most to the prediction of the STM. For illustration we grouped the variables as a function of their labels (i.e, pitch, intensity etc., see labels in Supplementary Table S2) but note that the complete list of VIP scores is presented in Figs. S4 and S5.

On the basis of these results, we proceeded to ask whether acoustical data alone are sufficient to classify speech and song vocalizations, or whether STM features are stronger predictors of vocalization type. Given the cross-cultural STM results and the theoretical motivation of this paper, we expected STM data to be *more* predictive of vocalization type (as opposed to STM and acoustic data being complementary, or the acoustic data outperforming the STM data). To test this question, we used SVM classifier with fieldsite-wise $k$-fold cross-validation (as in Fig. 2) to classify song and speech vocalization samples, using either (1) only the STM features, (2) STM features and acoustical data, (3) acoustical data only or (4) acoustical data only but without the VIP variables as input features.

All models were decoding speech and song above chance (Wilcoxon rank tests, two-tailed, W(20) values > 231 and all $p$-values < 0.001, Rank biserial correlation (effect size) = 1.00). However, the results showed a clear advantage of STM features over acoustic features: nonparametric RM ANOVA (Friedman) performed on decoding accuracy values revealed a main effect of models ($\chi^2$ (3) = 43.4, $p < 0.001$, Kendall's W coefficient (effect size) = 0.68), with post hoc tests (Durbin-Conover - Bonferroni corrected) showing significantly higher classification accuracy with STM features alone than with acoustical data alone ($p < 0.001$) and than acoustical data without VIP variables ($p < 0.001$). Moreover, classification performed with STM features and acoustical features was significantly more accurate than classification performed with acoustical data ($p < 0.001$), and than classification performed with acoustical data without VIP variables ($p < 0.001$). Additionally, classification performed with acoustical features was significantly more accurate than classification performed

with acoustical data without VIP variables ($p < 0.001$). Finally, the classification performed with STM features and the classification performed with STM features + acoustical features were not significantly different ($p = 0.91$). Spectro-temporal features of speech and song therefore more reliably distinguish between the vocalization types than do general acoustic features of the vocalizations.

Last, we asked whether human judgments could be linked to the acoustical features. To do so, we computed the correlation between SVM decoding accuracy of the acoustical data and the normalized difference between participants' categorizations of the vocalizations as speech and song, computed within each society. The decoding accuracy of the classifier trained on acoustical data was not correlated with the normalized behavioral scores ($r(20) = 0.14$, $p = 0.54$, Fig. S6), in sharp contrast with the STM data reported above (Fig. 4E). This suggests that participants prioritized STM features over acoustic features to inform their categorization decisions.

## Discussion

Using vocalizations drawn from a diverse set of languages and societies[6] we found that speech and song systematically differ in their typical acoustical signatures: songs contain greater energy than spoken utterances at higher spectral and lower temporal modulation rates, whereas speech shows the reverse effect (Fig. 2A). This pattern was sufficiently consistent that, despite variation in the distributions of STM patterns in the vocalizations of each society tested (Figs. S7–S10), we still observed near-complete overlap within each category (song and speech) in the two specific acoustical ranges across nearly all of the societies (Fig. 2B); conversely, there was essentially no overlap between the two categories, as shown by the white space between the blobs in Fig. 2B.

That these spectro-temporal cues suffice to classify the two categories well was shown by the outcome of a machine-learning classifier, which was trained exclusively on the spectro-temporal features, and correctly identified both classes of vocalization well above chance (mean 84.5% with comparable sensitivity and specificity) for all of the 21 societies (Fig. 2D), albeit with differing degrees of accuracy. To verify that this outcome was not merely driven by similarities in the speech or song samples across societies that may have been geographically or linguistically related, we tested the classifier using only data from one country/region or language family (classifier trained on the others); the outcomes were essentially the same (Fig. 3). Furthermore, the information used by the classifier (Fig. 2F) corresponded well to the ranges of modulation power that characterize the two classes, as identified in the initial univariate analysis (Fig. 2A), indicating that the classifier achieved high performance by using these same ranges of STM (see Supplementary Discussion and Figs. S11 and S12 for generalization of our findings to Cantonese, a tonal language with more than one level tone).

We also show that human listeners who were unfamiliar with most of the speech and song systems sampled here performed close to ceiling when asked to indicate which vocalization corresponded to which category (Fig. 4B). Their ratings were directly related to the distribution of energy in the modulation power spectrum (Fig. 4D), and the ranking of behavioral accuracy across societies was similar to that of the classification algorithm (Fig. 4E, see also Fig. S3), suggesting that both the classifier and the humans relied on the same STM cues that descriptively differentiate the two classes.

Finally, we showed that spectro-temporal modulation features are sufficient to differentiate speech and song vocalization at a high level of accuracy (~85%), because the addition of 98 other acoustical features (pitch, formants, intensity, vowel rates, harmonicity, rhythmic measures etc.) that are widely used to characterize speech and song did not improve performance. Classification performed with spectro-temporal features was also significantly more accurate than classification performed with these other acoustical variables, except insofar

as they also overlap with STM information (Fig. 5). This observation suggests that STM cues are necessary for good classification of the two categories. In fact, whereas behavioral performance of our listeners correlated well with STM-based classification, it did not correlate significantly with the classification using the other acoustical variables (Fig. S6). Thus, whether or not they are aware of it, listener categorizations of speech and song are apparently driven by the spectro-temporal characteristics of the vocalizations, not by other acoustic characteristics.

The findings support the idea that some universals exist in the acoustical manifestations of the two principal modes of cognitively complex auditory-vocal communication found in our species. Because the differences in specific ranges of STM patterns for speech and song are widely shared across culturally, linguistically, or geographically unrelated groups of people, we propose that they represent a fundamental property of how sounds are generated by the human vocal tract, depending on the nature of the communication. To transmit denotative information using speech, a high level of temporal modulation is used, but spectral modulation is less prominent; whereas to communicate musical content and affective states using song, a high level of spectral modulation is used, but at lower temporal modulation rates. The fact that people unfamiliar with the most of the linguistic or musical systems in question were nevertheless easily able to identify which vocalization belonged to which category, and that they used essentially the same spectro-temporal cues as the machine-learning classifier had determined to be optimal, supports the conclusion that such cues are widely shared and readily available even in the absence of any culturally specific knowledge.

One potential explanation for the distinct spectro-temporal signatures of speech and song is that they result from differences in neural control over the vocal musculature during speaking versus singing[14]. The higher temporal modulation in speech reflects the syllable rate (opening and closing of the mouth), which tends to be faster when speaking than when singing[31]. The longer syllable duration in singing may allow for production of more stable pitch values, leading to better encoding of tonal relationships important for music[45]. Conversely, the high spectral modulation rate associated with song may be related to the complex physiology of phonation typical of singing that generates more energy in the upper harmonics[13,46].

Most songs, including those used here, incorporate both spoken and melodic content simultaneously. Thus, both types of modulation are typically present together. But what distinguishes the two is their different acoustical signature, as determined by comparing them against one another (Fig. 2A and Figs. S7–S10). This is not to say that all cultures necessarily carve out the spectro-temporal space in exactly the same way. Indeed, although there was almost complete overlap of at least part of the distribution of STM patterns for both speech and song across societies (Fig. 2B), and the centroids of each distribution were clustered in close proximity (Fig. 2C), a glance at the individual modulation difference plots for each culture (Figs. S7–S10) reveals that there are important differences across them, especially in the songs, which exploit wide ranges of spectral and temporal modulation, even if they are generally fairly far from the range of modulations used for speech. Further study of how and why these cues are deployed in different musical traditions could help to identify and explain such cross-cultural differences. Indeed, the spectro-temporal framework may prove particularly valuable for examining questions of cross-cultural variability in language and music, since it does not require the selection of any particular linguistic or musical features, which are notoriously vulnerable to culture-specific assumptions[18]; see also Supplementary Information in ref. 8.

The clear acoustical distinction between song and speech should not be taken to imply that top-down factors have no influence on the perception of a vocalization as song or as speech. Indeed, the well-known "speech-to-song" illusion[47] demonstrates that speech may

sometimes be perceived as song after repeated presentation, even if the acoustics are held constant. This phenomenon has been attributed both to particular acoustical features of sounds susceptible to the illusion, as well as to individual differences across listeners[48–50]. The spectro-temporal framework may provide a useful approach to investigate vocalizations that are intermediate between canonical speech and song, and which may share features of both, not only in the context of the speech-to-song illusion, but also more broadly to study artistic forms in which speech and song features are blended (e.g. rap), or in speech with more prominent song-like features (e.g. infant-directed speech[6]).

The findings presented here fit well with previous empirical work examining the perceptual relevance of acoustical cues for speech and music. Several prior studies have shown that temporal rates of speech samples from different languages are in the range of 4–6 Hz[15,38]. Music from Western and from diverse cultures is generally less than half the speed of speech[12,15,38]. These observations are compatible with ours, in which speech temporal modulation occupied a range of 5–8 Hz, while song temporal modulations were close to 1 Hz. Other acoustical features have been proposed as distinguishing song from speech, including pitch height, harmonicity, rhythmic regularity, and discrete pitches[7,12,19]. Our analysis of 98 acoustical features however suggests that the STM framework provides a parsimonious way of accounting for the two categories, since STM features alone generated high classification rates, while the addition of the other features did not improve classification.

In a direct test of the importance of spectral and temporal cues for song and speech, a previous study[33] found that perception of English or French speech content in songs remained largely intact with spectral degradation, but quickly deteriorated with temporal degradation, whereas perception of the melodic content of the songs was largely abolished with spectral degradation but was not much affected by temporal degradation. Those findings built upon prior studies showing the importance of temporal modulations for English speech[32,41]. The current results extend the conclusions about the importance of temporal and spectral cues for speech and melody, respectively, beyond Western linguistic and musical systems, to encompass a widely distributed set of cultures.

The differences we observed in the present study for speech and song can be interpreted within the context of neuroscience findings that suggest partially dissociable neural representations of the two types of signals. Recent functional MRI data[51] using a voxel decomposition approach suggest that speech and music have distinct cortical representations as cognitive domains, rather than on the basis of acoustical cues. Indeed, intracranial recordings suggests the existence of a cortical region, located bilaterally within the anterior temporal lobes, that is specifically sensitive to song over all other sound categories[52]; interestingly, the same dataset also shows specific sensitivity to spectral and temporal modulation in different, peri-primary cortical regions.

A competing idea is that speech content vs song melody are processed in distinct auditory cortical regions as a function of hemispheric differences in sensitivity to spectral and temporal modulations[33]. Numerous studies have adduced evidence that the neuronal populations in left auditory cortex have higher temporal resolution but lower spectral resolution, whereas the right auditory cortex has the reverse specialization[36,53–55]. According to this view, speech and song are represented in distinct neural substrates not because of domain-specific aspects, but rather because of their tendency to utilize opposite ends of the spectro-temporal continuum. The data from the present study would be in line with this conclusion, insofar as the STM signatures of speech and song are shown to be sufficient to distinguish the two categories across many different linguistic and musical systems, suggesting that they reflect a fundamental organizational specialization of the human brain to process the two acoustical dimensions.

Our findings are therefore compatible with a biological origin of speech and song, upon which cultural influences act to produce the rich, varied, and beautiful forms of language and music found throughout the world. This possibiliy aligns well with three ideas about human auditory perception. First, it fits with the hypothesis of efficient coding, according to which the nervous system optimizes its representation of the environment based on the most salient features necessary for success[56]. Thus, neural responses are well-matched to the statistical properties of the most important aspects of both the visual[57] and auditory worlds[58] of a given species. Second, in auditory neuroscience, the STM framework has proven successful in accounting for the processing of complex environmental sounds, including vocalizations[24–28], in several species, which fits well with our conclusions. Third, it is well aligned with the hypothesis that music and speech tend to have distinct functional roles[2,39,40] in human evolution. Humans talk and sing, in part due to a nervous system that enables the generation and perception of signals that occupy different portions of the spectro-temporal continuum, allowing us to communicate the richness of our thoughts, ideas, and emotions with one another.

## Methods

This research complies with all relevant ethical regulations; the experimental procedures were approved by the Ethics Review Board of the CIUSSS de la Capitale Nationale (2022-2476).

### Vocalization corpus

We used a corpus of 738 recordings of adult-directed song, and adult-directed speech (all audio is available at https://doi.org/10.5281/zenodo.5525161) from[6]. People ($N = 369$) living in 21 societies produced each of these vocalizations, respectively, with a median of 15 individuals per society (range 6-57). From those for whom information was available, 86% were female.

Recordings were collected by the investigators of ref. 6 and/or staff at their fieldsites, all using the same data collection protocol. They translated instructions to the native language of the participants, following the standard research practices at each site. Fieldsites were selected partly by convenience (i.e., via recruiting principal investigators at fieldsites) and partly to maximize cultural, linguistic, and geographic diversity (see Table S1).

For speech recordings, participants spoke to the researcher about a topic of their choice (e.g., they described their daily routine). For song, participants sang a song that was not intended for infants (see ref. 6 for details); they also stated what that song was intended for (e.g., "a celebration song"). Participants vocalized in the primary language of their fieldsite, with a few exceptions (e.g., when singing songs without words; or in locations that used multiple languages, such as Turku, which included both Finnish and Swedish speakers).

Participants were free to determine the content of their vocalizations. This was intentional: imposing a specific content category on their vocalizations would likely alter the acoustic features of their vocalizations, which are known to be influenced by experimental contexts[6].

All recordings were made with Zoom H2n digital audio recorders, using foam windscreens (where available). To ensure that participants were audible along with researchers, who stated information about the participant and environment before and after the vocalizations, recordings were made with a 360° dual x−y microphone pattern. This produced two uncompressed stereo audio files (WAV) per participant at 44.1 kHz; we only analyzed audio from the two-channel file on which the participant was loudest.

The investigator at each fieldsite provided standardized background data on the behavior and cultural practices of the society (e.g.,

whether there was access to mobile-phones/TV/radio, and how commonly people used Infant Directed (ID) speech or song in their daily lives). Most items were based on variables included in the D-PLACE cross-cultural corpus[6]. The 21 societies varied widely in their characteristics, from cities with millions of residents (Beijing) to small-scale hunter-gatherer groups of as few as 35 people (Hadza). All of the small-scale societies studied had limited access to TV, radio, and the internet, mitigating against the influence of exposure to the music of other societies. Four of the small-scale societies (Nyangatom, Toposa, Sápara/Achuar, and Mbendjele) were completely without access to these communication technologies.

Our strategy was to use the longest possible segments available. To do so, for each speaker who produced both speech and song samples, we used whatever duration of each was available, always matching the durations within-speaker. For example, if a given speaker produced speech for 15 s and song for 12 s, we would take the first 12 s of the speech and compare it to the full 12 s sample for song.

### Extraction of spectro-temporal modulations

For the 738 selected samples (369 speech and 369 song) we decomposed the acoustical signal using the framework of spectro-temporal modulation power[41]. This analysis was done using the duration of the shorter sample (song or speech) produced by the same speaker. The modulation domain results from the 2D fast Fourier transform of the autocorrelation matrix of the sound stimulus in its spectrographic representation and represents the energy modulation across the temporal and spectral axes (Fig. 1). This results in 738 STM patterns data that were then used for univariate and multivariate analyses.

### Univariate analyses

Fieldtrip[42] functions as implemented in Brainstorm[43] were used to perform nonparametric permutation statistics with FDR correction ($p < 0.001$) for the contrast between song and speech STM patterns. Nonparametric tests were chosen as we did not make any assumption about the distribution of the STM data.

### Multivariate analyses

Multivariate analyses were performed using MATLAB and linear support vector machine (SVM) implementation (https://www.mathworks.com/help/stats/fitcecoc.html). A linear classifier was chosen as STM data contains many more features than examples, and classification of such data is generally susceptible to over-fitting. One way of alleviating the danger of over-fitting is to choose a simple function (such as a linear function) for classification, where each feature affects the prediction solely via its weight and without interaction with other features (rather than more complex classifiers, such as nonlinear SVMs or artificial neural networks, which can let interactions between features and nonlinear functions thereof drive the prediction). With small stimulus sets it is typically necessary to regularize SVM analyses. The regularization parameter (λ) serves as a degree of importance that is given to misclassifications. SVM pose a quadratic optimization problem that looks for maximizing the margin between both classes and minimizing the amount of misclassifications. Different values of λ will vary the misclassification constraint: when λ tends to infinite the solution tends to the hard-margin (allow no miss-classification). When λ tends to 0 the more the miss-classifications are allowed. Here the regularization parameter has been set to λ = 0.01 and has been selected using a separate validation set (infant-directed song ang speech from ref. 6, stimuli available here: https://zenodo.org/record/5525161). We performed the same analysis as reported above using SVM classifier with fieldsite-wise k-fold cross-validation to classify infant-directed song and speech vocalization samples, using the STM as features. Results were also expressed as accuracy of category identification that was calculated using an average of the cross-validation folds. The selection of λ was done as follows: on the training set, we estimate

several different models, with different values of the regularization parameter (λ = 0.1, λ = 0.05, λ = 0.01, λ = 0.005, λ = 0.001), then on the validation set, we choose the best model (the regularization parameter which gives the highest accuracy on the validation set).

Our strategy was to use the SVM classifier with fieldsite-wise k-fold cross-validation to classify song and speech vocalization samples, using the STM as features. The model is trained only on data from 20 of the 21 societies to predict whether each vocalization in the 21st society is song or speech. The procedure is repeated 21 further times, with each society being held out, to estimate the classification performance across the full set of societies. Results were expressed as accuracy of category identification that was calculated using an average of the cross-validation folds. For each classifier, we extracted the features weights (z-score) to evaluate the relative contribution of each feature in the classification. This procedure was performed across societies (21), across countries (18), language families (16), world subregions (15) and regions (6).

### Behavioral experiment

**Participants.** 80 adults participated in the behavioral experiment. No statistical method was used to predetermine sample size. The group was composed of 80 native French speakers from France and Canada (33 female, 4 non-binary, mean age = 32.4 years ± 10.86). Some of them (10 out of 80) were musically trained (more than 5 years of formal musical training). Six participants did not complete the entire test and the data of 74 participants were included in the current study. Participants reported no history of neurological or psychiatric disease. All participants provided written informed consent, and the experimental procedures were approved by the Ethics Review Board of the CIUSSS de la Capitale Nationale (2022–2476). The study has been conducted according to the principles expressed in the Declaration of Helsinki.

**Procedure.** The experiments took place in a sound-attenuated booth. Auditory stimuli were presented binaurally via Sennheiser HD 280 pro headphones at a comfortable sound level (~75 dB SPL). PsychoPy[59] was used to control the stimulus presentation and record responses. We played the song and speech recordings to these individuals who were asked to rate, as rapidly as possible on a 5-point scale on their keyboard, whether each speaker was singing (code 1) or speaking (code - 1). Participants had 9 s to respond and received no feedback (i.e., we did not tell them whether or not their rating was accurate). We did not provide any criteria to the listeners; they judged the sounds based on whatever they thought was relevant (hence the minimal instructions) so as to avoid any kind of bias about what features to use, hence providing a clean test of whether they would spontaneously use similar features as the classifier did. The experiment lasted approximately 15 min. We used 3 different blocks that were pseudo-randomly presented to the participant. Each bloc contained the same number of examples of speech and song for each society, with a total of 246 trials per block. This way a given listener was also rating the vocalization of the same speaker. Example of this task can be found online: https://run.pavlovia.org/palbouy/spectrotemp_bloc1.

**Behavioral data analysis.** Data were processed with MATLAB (The Mathworks), and statistical analyses were performed with Jamovi (https://www.jamovi.org). For each participant, the ratings corresponding to scores in a linear scale (singing (code 1) to speaking (code −1), see Fig. 4) were extracted and averaged for each participant separately for each society, language family, countries, world subregions and world regions. These scores were analyzed with nonparametric tests (two-sided) and we performed Pearson's correlation between behavioral scores and decoding accuracy/raw spectro-

temporal patterns that were corrected with FDR ($p < 0.05$) when necessary.

## Analysis of acoustical data

Acoustical analyses were done on a broad range of acoustical variables extracted in ref. 6– and can be found here https://github.com/themusiclab/infant-speech-song/tree/main/data. All details about the extraction of these acoustical variables are described in the original article. These acoustical variables are summarized in the Table S2 (table from ref. 6, used with permission).

We first aimed to investigate the link between acoustical features and STM data. To do so performed a Partial Least Squares (PLS) analysis as implemented in MATLAB https://www.mathworks.com/help/stats/plsregress.html using acoustical data as predictor variables and STM data as response variables. We used PLS instead of multiple linear regression models as multicollinearity existed between several variables of the acoustical dataset. PLS analysis was done for each spectral/temporal coordinate of the entire STM domain across all sounds (song and speech) resulting in one PLS model per spectral/temporal coordinate. For each model we then extracted the fitted responses values and estimated if the models could significantly fit the STM data. To do so, we computed the correlation (FDR corrected, $p < 0.05$) between the fitted responses (PLS models) and observed response (STM features) in the entire STM domain.

To determine which acoustical variables contributed more to the prediction, we calculated the variable importance in projection (VIP) scores for the PLS regression models for spectral and temporal statistical peaks reported in Fig. 5A. A VIP score is a measure of a variable's importance in the PLS model. In other words, it summarizes the contribution a variable makes to the model. The VIP score of a variable is calculated as a weighted sum of the squared correlations between the PLS components and the original variable. The weights correspond to the percentage variation explained by the PLS component in the model. As variables with a VIP score greater than 1 are considered important for the projection of the PLS regression model[44], we report in the main text only variables that were above this threshold. To facilitate illustration, we grouped variables according to their labels (see Table S2 for the corresponding labels) but present the complete list in Figs. S4 and S5.

Moreover, to investigate whether spectro-temporal modulation features were superior to classical acoustical features to differentiate speech and song vocalization, we used SVM classifier with fieldsite-wise k-fold cross-validation to classify song and speech vocalization samples, using either (1) only the STM features, (2) STM features and acoustical data, (3) acoustical data only or (4) acoustical data only but without the VIP variables as input features and compared model accuracy using nonparametric RM ANOVA (Friedman).

Finally, to investigate whether human judgments were linked to the acoustical features we computed Pearson's correlation between the SVM decoding accuracy of the model using acoustical data only as features and the normalized difference between Song and Speech behavioral ratings computed within each society.

## Reporting summary

Further information on research design is available in the Nature Portfolio Reporting Summary linked to this article.

## Data availability

The data generated in this study have been deposited in an OSF database. Raw vocalizations and acoustical data are freely available at https://zenodo.org/record/5525161 and https://github.com/themusiclab/infant-speech-song/tree/main/data[6]. The processed data are available at: https://doi.org/10.17605/OSF.IO/XCSQM[60]. Example of the judgment task can be found here:: https://run.pavlovia.org/palbouy/spectrotemp_bloc1.

## Code availability

MATLAB codes are freely available at the following https://doi.org/10.17605/OSF.IO/XCSQM[60].

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

## Acknowledgements

This work was supported via a CIHR Foundation grant to R.J.Z. and an NSERC Discovery grant to P.A. (RGPIN-2019-06162). R.J.Z. is a fellow of the Canadian Institute for Advanced Research and is funded via the Canada Research Chair program, and by the Scientific Grand Prize from the Fondation pour l'Audition (Paris, France). P.A. was supported by FRQS Junior 1 (280380) and Junior 2 (329968) grants. S.A.M. is supported by grants from the US National Institutes of Health (DP5OD024566) and the Royal Society of New Zealand Te Apārangi (RDF-UOA2103 and MFP-UOA2133).

## Author contributions

Conceptualization, R.J.Z., P.A., and S.A.M.; Methodology, P.A., S.A.M., Y.D., and R.J.Z.; Analysis, P.A.; Investigation, P.A., S.A.M., R.S.H., J.G., and Y.D.; Resources, P.A. and R.J.Z.; Writing – Original Draft, R.J.Z. and P.A.; Writing – Review & Editing: P.A., S.A.M., R.S.H, J.G., Y.D., and R.J.Z; Visualization, P.A.; Supervision and Project Administration, R.J.Z., P.A., and S.A.M.

## Competing interests

The authors declare no competing interests.
