## [Peer Review File · Nature Communications]

Spectro-temporal acoustical markers differentiate speech from song across culturesEditorial Note: This manuscript has been previously reviewed at another journal that is not operating a transparent peer review scheme. This document only contains reviewer comments and rebuttal letters for versions considered at *Nature Communications* .

REVIEWERS' COMMENTS

Reviewer #1 (Remarks to the Author):

The authors considered all of my reviews, revised sufficiently. Thank you!

Reviewer #2 (Remarks to the Author):

I am happy with the authors' responses to my concerns; in particular, I agree that the incorporation of sample duration into the analysis suggests that some of the limitations I pointed out in my original review are driven by this factor rather than cross-cultural variation. My one lingering suggestion would be that I think it would be worth incorporating into the Supplementary Information the data on Cantonese excerpts included in the response to reviewers. I understand the argument behind leaving it out of the manuscript, which is otherwise based on a dataset in which speech and song are produced by the same speakers. Nevertheless, I think this analysis of the Cantonese excerpts provides important preliminary evidence that the findings may generalize to languages with more than one level tone, and therefore that it would be best to make it accessible, even if it is left out of the paper's main narrative.

We thank the reviewers for their positive evaluation of our study. You will find our response to reviewer 2 below:

Reviewer #2 (Remarks to the Author):

I am happy with the authors' responses to my concerns; in particular, I agree that the incorporation of sample duration into the analysis suggests that some of the limitations I pointed out in my original review are driven by this factor rather than cross-cultural variation. My one lingering suggestion would be that I think it would be worth incorporating into the Supplementary Information the data on Cantonese excerpts included in the response to reviewers. I understand the argument behind leaving it out of the manuscript, which is otherwise based on a dataset in which speech and song are produced by the same speakers. Nevertheless, I think this analysis of the Cantonese excerpts provides important preliminary evidence that the findings may generalize to languages with more than one level tone, and therefore that it would be best to make it accessible, even if it is left out of the paper's main narrative.

We have added the following information (see page 9 or below) in the main text and a new section in the supplementary information file (pages 12 to 14).

Main text, Page 9: "Furthermore, the information used by the classifier (Fig. 2F) corresponded well to the ranges of modulation power that characterize the two classes, as identified in the initial univariate analysis (Fig. 2A), indicating that the classifier achieved high performance by using these same ranges of STM (see supplementary discussion and Figs S11 and S12 for generalization of our findings to Cantonese, a tonal language with more than one level tone)."

Supplementary information, Pages 12 to 14):

Supplementary Discussion

It is relevant to note that the phenomenon of lexical tone should be considered in the current study. Indeed, in languages with lexical tone, the discreteness of pitch variation in speech and song is more similar compared to languages without lexical tone. In the Hilton and al. database only two of the twenty-one societies included featured languages with lexical tone, and did not include languages with multiple level tones, such as Cantonese. In order to address this limitation we carried out both univariate and multivariate analyses on Cantonese song samples (20 samples recorded in Beijing) vs. Cantonese speech (20 sample from the Cantonese ASR database: <https://github.com/HLTCHKUST/cantonese-asr/tree/main/dataset> matched in gender and duration with the song samples, see Fig. S11 and S12 below).

Fig. S11. Extraction of spectro-temporal modulations patterns for singing and speaking vocalization samples in the Cantonese sample. We extracted the spectrotemporal modulation patterns of the Cantonese sample and then averaged them. We computed the normalized difference between the song and speech modulation patterns. The panel in the third column therefore show where in the modulation space song differs from speech (red part of the scale) and where speech differs from song (blue part of the scale).

Univariate analysis (Song vs. Speech contrast of STM) revealed a clear overlap between the Cantonese analysis and the analysis performed with the 21 societies, within each category (song and speech) in the two specific acoustical ranges reported in the main text (see Fig. 2. and Figure S7 to S10). That is, the orange zones in the MPS, corresponding to energy found to be greater in song than speech largely fell within the range observed in the other 21 groups (black outlines from Fig. 2A). Similarly, the blue zones, corresponding to greater energy for speech compared to song, fall well within the range for speech in the univariate analysis from Fig 2A. So descriptively speaking, Cantonese is not different from what we expected.

Fig. S12. Song vs. Speech contrast in the STM domain for Cantonese excerpts ($p < .05$, FDR-corrected). Dark lines illustrate the boundaries of the significant effects presented in Fig. 2A of the manuscript.

For multivariate results, we trained a classifier exclusively on the spectro-temporal features of the 21 societies reported in the manuscript (see methods) and then made predictions on Cantonese vocalizations. The model successfully classified Cantonese song and speech well above chance (accuracy = 80%; sensitivity = 85%, specificity = 75%). Thus, Cantonese speech and song share sufficient commonalities in terms of their spectrotemporal modulations with other societies that the model derived from those other societies generalizes quite well.

Overall, these results suggest the results reported in the main text can generalize even to a language with multiple lexical tones. There is one caveat, which is that we were unable to find a dataset containing Cantonese speech and song produced by the same speakers. From the samples we had available, we matched speech and song in terms of the gender of the speaker, but there are likely to be residual acoustical differences due to variable vocal characteristics. This effect, which we cannot easily quantify, would most likely add variability to our estimates of the STM of speech and music, and so the data we present above are probably a conservative estimate of the degree to which Cantonese is similar to the other groups. That is, if we had a nicely matched set of vocalizations, we expect the result would be even better than it is.